# Metabolic-Syndrome-Related Comorbidities in Narcolepsy Spectrum Disorders: A Preliminary Cross-Sectional Study in Japan

**DOI:** 10.3390/ijerph19106285

**Published:** 2022-05-22

**Authors:** Kunihiro Futenma, Yoshikazu Takaesu, Masaki Nakamura, Kenichi Hayashida, Noboru Takeuchi, Yuichi Inoue

**Affiliations:** 1Department of Neuropsychiatry, Graduate School of Medicine, University of the Ryukyus, 207, Uehara, Nishihara-cho, Nakagami-gun, Okinawa 903-0215, Japan; takaesuy@med.u-ryukyu.ac.jp; 2Japan Somnology Center, Neuropsychiatric Research Institute, 5-10-10, Yoyogi, Shinjuku-ku, Tokyo 151-0053, Japan; m-nakamura@omotesando-sleep.com (M.N.); takeuchi@somnology.com (N.T.); 3Department of Somnology, Tokyo Medical University, 6-7-1, Nishishinjuku, Shinjuku-ku, Tokyo 160-0023, Japan; 4Aoyama-Omotesando Sleep Stress Clinic, Aoyama Rise Square 3F, 5-1-22 Minamiaoyama, Minato-ku, Tokyo 107-0062, Japan; 5Sleep Support Clinic, Miranbeena 1F, 1-18-8 Higashioi, Shinagawa-Ku, Tokyo 140-0011, Japan; hayashida@sleep-stress.com; 6Department of Neuropsychiatry, Kurume University School of Medicine, 67 Asahi-machi, Kurume 830-0011, Japan

**Keywords:** narcolepsy spectrum disorders, narcolepsy type 2, metabolic syndrome, orexin, HLA-DQB1*06:02, obesity

## Abstract

Narcolepsy types 1 (NT1) and 2 (NT2) and idiopathic hypersomnia (IH) are thought to be a disease continuum known as narcolepsy spectrum disorders (NSDs). This study aimed to assess the prevalence of and factors associated with metabolic-syndrome-related disorders (MRDs) among patients with NSD. Japanese patients with NSD (NT1, *n* = 94; NT2, *n* = 83; and IH, *n* = 57) aged ≥35 years were enrolled in this cross-sectional study. MRD was defined as having at least one of the following conditions: hypertension, diabetes, or dyslipidemia. Demographic variables and MRD incidence were compared among patients in the respective NSD categories. Multivariate logistic regression analysis was used to investigate the factors associated with MRDs. Patients with NT1 had a higher body mass index (BMI) and incidence of MRD than that had by those with NT2 or IH. Age, BMI, and the presence of OSA were significantly associated with the incidence of MRD in NSDs. Age and BMI in NT1, BMI and human leukocyte antigen (HLA)-DQB1*06:02 positivity in NT2, and only age in IH were factors associated with the incidence of MRD. Obesity should be carefully monitored in narcolepsy; however, NT2 with HLA-DQB1*06:02 positive should be followed up for the development of MRD even without obesity.

## 1. Introduction

Narcolepsy is a disorder characterized by excessive daytime sleepiness (EDS) and increased rapid eye movement (REM) sleep propensity, resulting in cataplexy, sleep paralysis, and hypnagogic hallucinations [1,2]. The International Classification of Sleep Disorders, Third Edition (ICSD-3) classifies the disorder into narcolepsy types 1 (NT1) and 2 (NT2) [3]. Idiopathic hypersomnia (IH) was formerly classified as IH with or without long sleep time (LST) in the ICSD, Second Edition (ICSD-2) [4]; however, IH with and without LST were merged into a single IH category in the ICSD-3 [3]. NT1, NT2, and IH without LST have been suggested to be part of a disease continuum known as narcolepsy spectrum disorders (NSDs) [5,6,7,8]. Deficiency in orexin (hypocretin), a neuropeptide in the neurons, is the main cause of NT1 [9]. Little is known about the etiology of NT2, but NT2 is believed to share some common features with NT1 in terms of EDS and increased REM sleep propensity. However, patients with NT2 are present without cataplexy or orexin secretion deficiency in the cerebrospinal fluid (CSF) [10]. Moreover, it has been widely accepted that dysfunction of the orexinergic system is significantly associated with human leukocyte antigen (HLA)-DQB1*06:02 positive narcolepsy [11]. Most patients with narcolepsy who have low levels of orexin in the CSF (<110 pg/mL) are HLA-DQB1*06:02 positive (97%) and cataplectic (98%) [12].

In general, NT1 is likely to be complicated by metabolic diseases, including obesity, diabetes, dyslipidemia, and hypertension [13,14,15]. Higher mortality rates have also been reported among elderly patients (age, >60 years) with narcolepsy [13,14]. However, it remains unclear whether NT2 and IH increase the risk of mortality or metabolic disease-related comorbidities as observed with NT1. Moreover, no study has investigated the risks of obesity, metabolic-disease-related comorbidities, and mortality associated with NT2 in consideration of HLA positivity/negativity. This study aimed to assess the incidence of medical complications related to metabolic syndrome and the associated factors among Japanese patients with NSD, particularly NT2, considering the presence/absence of HLA-DQB1*06:02 using a preliminary cross-sectional survey. Our main hypothesis is shown in Figure 1.

## 2. Materials and Methods

### 2.1. Surveyed Participants and Procedures

The present study was initiated after approval by the ethics committee of the Neuropsychiatric Research Institute (Tokyo, Japan), which is a Public Interest Incorporated Foundation licensed by the Japanese Cabinet Office (Approval number 143). Written informed consent was obtained from all participants before the initiation of the study. Consecutive Japanese outpatients with NSD (NT1, *n* = 94; NT2, *n* = 83; and IH, *n* = 57) who visited three clinics specializing in sleep disorders in Tokyo, Japan, between September 2017 and December 2019 were enrolled in this study. Considering the higher peak age of metabolic disease onset among the general population [15,16], we excluded patients aged <35 years. The diagnoses of NT1, NT2, and IH were made according to the ICSD-3 criteria based on clinical interviews by board-certified physicians specializing in sleep medicine and the data of polysomnography (PSG) and multiple sleep latency tests (MSLT) [3]. Sleep hygiene instruction was given prior to PSG and MSLT, with instructions to sleep longer (at least 7 h per day) and to record a sleep diary for approximately one month to exclude insufficient sleep syndrome and circadian rhythm sleep–wake disorders. All patients with NT1 were confirmed to have cataplexy symptoms, and no patient with NT2 had symptoms indicative of cataplexy. All of the eligible patients with NSD were regularly taking psychostimulants (modafinil, 100–300 mg; methylphenidate hydrochloride, 10–60 mg and/or pemoline, 10–200 mg). Clomipramine hydrochloride (10–40 mg) was used by 91.0% of eligible patients with NT1, by 6.0% of those with NT2, and by none of those with IH.

HLA-DQB1*06:02 typing was performed for all patients with NT2 to classify them into two categories including patients with NT2 who were HLA-DQB1*06:02 positive (NT2P, *n* = 36) and who were HLA-DQB1*06:02 negative (NT2N, *n* = 47). Sixty-three patients with NT1 underwent HLA-DQB1*06:02 typing, and most of them were HLA-positive (positivity rate: 95.2%, 60/63). HLA typing was not performed in patients with IH. To focus exclusively on NSD in this study, we also did not consider IH with LST (more than 11 h of sleep per day).

Demographic data, including age, sex, and body mass index (BMI), were collected. Past/current medical information was investigated using self-completed questionnaires and inquiries by physicians. To ensure the accuracy of the diagnoses of complications, blood pressure was checked more than once at different time points during the survey period, and blood tests, including biochemical examinations, were conducted.

In this study, we provisionally defined a patient with a metabolic-syndrome-related disorder (MRD) as a patient with at least one of the following conditions: hypertension, diabetes, or dyslipidemia. Hypertension was defined as systolic blood pressure ≥140 mmHg or diastolic blood pressure ≥90 mmHg according to the Japanese Society of Hypertension Guidelines for the Management of Hypertension [17]. A participant was diagnosed with diabetes if the hemoglobin A1c (National Glycohemoglobin Standardization Program) level was ≥6.5%, according to the diagnostic criteria for diabetes mellitus in Japan [18]. Dyslipidemia was diagnosed as fasting low-density lipoprotein cholesterol ≥140 mg/dL, fasting high-density lipoprotein cholesterol ≤ 40 mg/dL, or fasting triglyceride ≥150 mg/dL according to the Japan Atherosclerosis Society Guidelines for Prevention of Atherosclerotic Cardiovascular Diseases 2017 [19]. Patients receiving medications for hypertension, diabetes, or dyslipidemia were considered positive for MRD, even if the clinical numerical values of these disorders were within normal limits at the time of the investigation. Obesity was defined as a BMI ≥ 25 kg/m^2^ according to the diagnostic criteria of the Japan Society for the Study of Obesity [20].

We also used respiratory disorder variables obtained from the diagnostic PSGs of participants with obstructive sleep apnea (OSA) for the analyses. In the ICSD-3, the criterion for OSA is an apnea–hypopnea index (AHI) ≥ 15/h (moderate to severe OSA) or an AHI ≥ 5/h together with self-reported symptoms, such as EDS [3]. Patients with moderate to severe OSA were selected for the analyses in this study, considering that all the participants with NSD had symptoms of EDS irrespective of whether they had OSA or not. Among the patients with moderate to severe OSA (AHI ≥ 15/h, *n* = 35), 24 were treated with nasal continuous positive airway pressure (nCPAP), 7 were treated using an oral appliance, and 4 did not receive any kind of treatment. All nCPAP-treated patients failed to meet the criteria for sufficient adherence, defined as at least 4 h of nCPAP treatment on 70% of the days monitored [21,22].

### 2.2. Statistical Analyses

One-way analysis of variance (ANOVA) was used to compare age and BMI among the three groups (NT1, NT2, and IH). Furthermore, post hoc Bonferroni analyses were performed when significant main effects were found. To compare nominal variables (sex and presence/absence of obesity, hypertension, diabetes, dyslipidemia, MRD, and moderate to severe OSA) among the three groups, a 2 × 3 chi-square test was used, followed by residual analyses.

In the next phase targeting NT2, we investigated the differences in demographic variables and the incidences of obesity, hypertension, diabetes, dyslipidemia, and MRD between patients who were NT2P and those who were NT2N. The independent *t*-test was used to compare numerical variables, the 2 × 2 chi-square test was used for nominal variables, and Fisher’s exact test was used when expected counts were less than 5.

To investigate the factors associated with MRD, we used multivariate logistic regression analysis with the presence or absence of an MRD as a dependent variable and the following as independent variables: age, sex, BMI, category of NSD (NT1, NT2, and IH), and the presence or absence of moderate to severe OSA (AHI ≥ 15/h). We used forced entry methods as the model of logistic regression, focusing on variables of clinical significance. Similar analyses were conducted for each category of NSD (NT1, NT2, and IH). In the analysis of patients with NT2, the HLA-DQB1*06:02 positive/negative status was also considered as an independent variable along with the aforementioned variables.

All statistical analyses were performed with EZR (Saitama Medical Center, Jichi Medical University, Saitama, Japan), a graphical user interface for R 2.13.0 (R Foundation for Statistical Computing, Vienna, Austria) [23]. More precisely, EZR is a modified version of R commander (version 1.6-3) designed to add statistical functions used frequently in biostatistics.

## 3. Results

### 3.1. Comparison of Demographic Variables and the Incidence of Complications among Patients with NSD

Table 1 shows a comparison of demographic variables among the patients with NT1, NT2, and IH. Age differed significantly among the three NSD categories (*p* < 0.01, one-way ANOVA). Post hoc Bonferroni analysis showed that the mean age of patients with NT1 was significantly higher than that of patients with NT2 (*p* = 0.013) and IH (*p* = 0.019); however, there was no significant difference in age between patients with NT2 and IH. There were significant differences (*p* < 0.01, one-way ANOVA) in mean BMIs between patients with NT1, NT2, and IH. Subsequent post hoc Bonferroni analysis revealed that the mean BMI of patients with NT1 was significantly higher than that of patients with NT2 (*p* < 0.01) and IH (*p* < 0.01). However, there was no significant difference in BMI between patients with NT2 and those with IH. A chi-square test for the incidence of pathological obesity (BMI ≥ 25) showed significant differences among the groups (*p* < 0.01), with adjusted residuals of 7.41 (NT1) and −2.87 (NT2). No significant difference in sex (*p* = 0.65, chi-square test) or the incidence of moderate to severe OSA with an AHI ≥ 15/h (*p* = 0.64, chi-square test) was observed among the three groups.

Table 2 shows a comparison of the incidence of complications among the three NSD groups. The incidence of hypertension differed significantly (*p* < 0.01) among the three groups and was highest and lowest among patients with NT1 (adjusted residual, 3.80) and NT2 (adjusted residual, −2.37), respectively. Meanwhile, there was no significant difference in the incidence of diabetes, although the margin was narrow (*p* = 0.05). There was also no significant difference in the incidence of dyslipidemia (*p* = 0.36) among the three NSD groups. Overall, there were significant differences in the proportions of individuals with any MRD among the three NSD groups (*p* = 0.04); however, the absolute value of the adjusted residual of the chi-square components was less than 1.96 (NT1 with an MRD, 1.85) (Table 2).

### 3.2. Comparison of Demographic Variables and Incidence of Complications between Patients Who Are NT2P and NT2N

The positivity rate for HLA-DQB1*06:02 was 43.4% among patients with NT2 (NT2P, *n* = 36 and NT2N, *n* = 47). The NT2P group had significantly fewer male patients (*p* < 0.01, chi-square test; adjusted residual, −2.18) than that had by the NT2N group. There was no significant difference (*p* = 0.89, *t*-test) in the mean age between patients who are NT2P and NT2N. There were no significate differences (*p* = 0.64, *t*-test) in mean BMIs between patients who were NT2P and NT2N. The incidences of obesity and moderate to severe OSA did not differ significantly (*p* = 0.41, chi-square test and *p* = 0.74, Fisher’s exact test, respectively; Table 1) between the NT2P and NT2N groups.

When the incidence of target complications was compared between patients in the two groups, the incidences of hypertension, dyslipidemia, and any MRD were higher among patients who were NT2P than among those who were NT2N (hypertension, *p* = 0.047 and adjusted residual, 2.09; dyslipidemia, *p* < 0.01 and adjusted residual, 4.29; and MRD, *p* = 0.019 and adjusted residual, 2.26). The incidence of diabetes did not differ significantly between the two groups (*p* = 0.09, Fisher’s exact test; Table 2).

### 3.3. Factors Associated with the Incidence of MRD

Multiple logistic regression analysis revealed that age (*p* < 0.01), BMI (*p* < 0.01), and presence of moderate to severe OSA (*p* < 0.05) were significantly associated with the incidence of MRD in the patients with NSD (Table 3).

Among the patients in each category of NSD, the incidence of MRD was significantly associated with age (*p* < 0.01) and BMI (*p* = 0.037) in NT1 (*n* = 94) (Table 4). Among patients with NT2 (*n* = 83), BMI (*p* = 0.012) and HLA positivity (*p* = 0.016) were significantly associated with the incidence of MRD (Table 5), and among patients with IH (*n* = 57), only age (*p* = 0.033) was significantly associated with the incidence of MRD (Table 6).

## 4. Discussion

The results of the present study showed that patients with NT1 had a higher BMI and incidence of MRD than those had by patients with NT2 or IH. Age, BMI, and presence of moderate to severe OSA were significantly associated with the incidence of MRD in NSD. Age and BMI were also significantly associated with the incidence of MRD among patients with NT1. However, only age was significantly associated with the incidence of MRD among patients with IH. The most novel finding of this study was that not only BMI but also HLA-DQB1*06:02 positivity was significantly associated with the incidence of MRD, independent of age and sex, among patients with NT2.

Obesity is one of the main causes of metabolic syndrome in the general population [24] and is commonly observed in patients with narcolepsy [25]. In particular, the onset of NT1 is likely to lead to rapid weight gain, which could increase to the level of pathological obesity at a relatively young age [26]. A previous study also suggested that the severity of obesity could be associated with the level of CSF-orexin deficiency in narcolepsy [27]. Therefore, confounding of NT1 with obesity may contribute to the result of this study that diagnostic classification of NT1 was not significantly associated with the incidence of MRD in NSD. A decreased basal metabolic rate due to orexin deficiency is believed to play an important role in the development of obesity, especially in NT1 [28]. On the contrary, studies on the basal metabolic rate have also yielded conflicting results [29]. The mechanism by which orexin deficiency contributes to the development of obesity remains unclear.

The results of the present study showed that hypertension was the most frequent MRD among patients with NT1. The 48.9% incidence of hypertension among patients with NT1 (53.6% and 42.1% among male and female patients, respectively) was higher than that in the general adult population, estimated to be 36.2% and 26.0% among men and women, respectively, according to the 2018 annual National Health and Nutrition Survey in Japan [30]. This finding is consistent with the results of an earlier study that showed that cardiovascular diseases, including hypertension, are more common among patients with narcolepsy than among the general population [31,32]. In another study, orexin knockout mice were shown to have a lower awake blood pressure than that had by wild-type mice [33]. In contrast, a study reported that orexin knockout mice had higher asleep blood pressure than that had by wild-type mice [34]. In addition, the proportion of non-dippers (defined as having a nocturnal diastolic blood pressure dip <10% lower than daytime blood pressure) was found to be significantly higher among patients with narcolepsy-cataplexy than among controls [35]. These findings support the idea that patients with orexin-lowered NT1 may be at risk of developing hypertension along with obesity.

OSA is a risk factor for the development of metabolic syndrome independent of obesity in the general population [36,37]. In addition, OSA causes a non-dipping nocturnal blood pressure pattern, which may lead to increased cardiovascular risk [38]. In this study, as in the general population, moderate to severe OSA was significantly associated with the incidence of MRD independent of obesity in patients with NSD. The nCPAP adherence of the patients with narcolepsy in this study was poor, possibly due to a lack of subjective improvement in sleepiness with the use of nCPAP. However, adequate treatment for OSA along with obesity prevention would also be desirable for preventing or improving MRD in patients with narcolepsy complicated by OSA.

Among the patients with NT2, our results indicated that patients who were HLA-DQB1*06:02 positive had a significantly higher incidence of hypertension and dyslipidemia than that had by those who were HLA-DQB1*06:02 negative. The HLA positivity rate among patients with NT2 in the present study was 43.4%, which is comparable to that reported in previous studies [3,39]. Interestingly, the present study showed that HLA-DQB1*06:02 positivity was significantly associated with the incidence of MRD, independent of age, sex, and BMI among patients with NT2. In this study, the CSF orexin level was not measured. A previous study indicated that approximately one-fourth of patients with narcolepsy without cataplexy studied in sleep centers specialized in narcolepsy across the world have orexin deficiency [40]. However, the loss of orexin tone was reported to be more partial in patients with CSF orexin lowered narcolepsy without cataplexy than in those with cataplexy, and compared to the previously established cutoff of 110 pg/mL, a diagnostic cutoff of 200 pg/mL could be more appropriate for these patients. [40]. A recent study reported that narcolepsy with intermediate CSF orexin levels (110–200 pg/mL) is a rare condition with a heterogeneous phenotype in which complications of cataplexy or obesity are less common compared to that observed in NT1 with CSF orexin levels below 110 pg/mL [41]. As mentioned above, the severity of obesity could be associated with the level of CSF orexin level in narcolepsy [27]. In this study, obesity was less common in patients with NT2 compared to patients with NT1; however, the result of this study might indicate that NT2 with HLA positivity requires more attention since even intermediately lowered CSF orexin levels possibly contribute to the development of MRD such as hypertension or dyslipidemia independent of obesity.

As observed in the general population, age was associated with the presence of MRD among patients with IH. Considering the normal CSF orexin levels among patients with IH [12], the risk of developing metabolic syndrome may be lower in patients with IH than in patients with orexin-deficient narcolepsy.

This study has several limitations. First, it was conducted as a cross-sectional survey, and a causal relationship between dependent and independent variables could not be confirmed. Second, since this preliminary survey was conducted in only three sleep disorder clinics in Tokyo, Japan, the study sample was small; therefore, the findings may not be representative of all patients with NSD. The most novel finding of this study was that HLA-DQB1*06:02 positivity was significantly associated with the incidence of MRD among patients with NT2; however, the power (calculated to be 0.651) was a little bit insufficient. Larger sample sizes would be desirable in future studies. Third, as mentioned above, we did not measure the CSF orexin levels of the participants; therefore, it is possible that some HLA-positive patients diagnosed with NT2 may turn out to be NT1 without cataplexy or NT2 with intermediately lowered CSF orexin levels if CSF orexin were measured. Further studies are needed to clarify the relationship between MRD markers and CSF orexin levels. Fourth, we could not compare the incidence of MRD among patients with NSD with that in the general population or among patients with IH with LST as controls. Fifth, we did not assess the smoking [42] or alcohol [43] status of the participants, although both are risk factors for metabolic syndrome. Likewise, the dose of psychostimulants should be considered, especially when estimating the possibility of developing hypertension in patients with NSD [44]. Sixth, as we described in the method, we excluded patients aged <35 years. However, CSF-orexin deficiency may cause metabolic syndrome at a relatively young age. Further studies of this kind should include younger patients. Seventh, as we mentioned in the result, there was no significant difference in the incidence of diabetes by a narrow margin (*p* = 0.05) among the three NSD groups. However, abnormalities of glucose metabolism may appear before reaching the clinical stage of diabetes. Given this, it would be effective to evaluate not only hemoglobin A1c but also insulin resistance (such as HOMA index) in further studies. Finally, in this study, we targeted MRD rather than metabolic syndrome itself. The National Institute of Health guidelines defines metabolic syndrome as having three or more of the following characteristics: (1) large waistline, (2) high triglyceride level, (3) low HDL cholesterol level, (4) high blood pressure, and (5) high fasting blood sugar [45]. However, hypertension, diabetes, and dyslipidemia may be the core components of metabolic syndrome. Considering these relationships, we defined MRD as having at least one of these three conditions.

## 5. Conclusions

The current study showed that patients with NT1 had a higher BMI and incidence of MRD than those had by patients with NT2 or IH. Age, BMI, and presence of moderate to severe OSA were significantly associated with the incidence of MRD in NSD. Age was the only significant factor associated with the incidence of MRD among patients with IH. However, BMI was significantly associated with the incidence of MRD among patients with NT1 and NT2. Moreover, HLA-DQB1*06:02 positivity was significantly associated with the incidence of MRD, independent of age, sex, and BMI, among patients with NT2. Thus, obesity is less common in patients with NT2 compared to those with NT1; however, patients with NT2 who are HLA-DQB1*06:02 positive should be carefully monitored for the development of MRD irrespective of the presence/absence of obesity. Further studies with larger sample sizes, including younger patients and general population controls, and prospective follow-up will be needed to confirm our results as well as to investigate the CSF orexin levels among patients with NT2.

## Figures and Tables

**Figure 1 ijerph-19-06285-f001:**
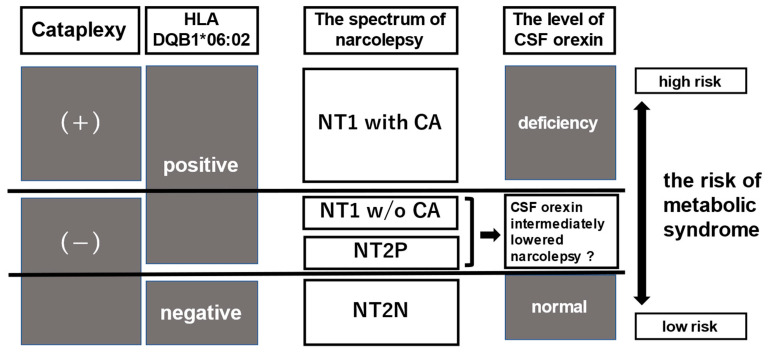
Narcolepsy, especially NT1, is considered to be an autoimmune disease triggered by a certain infection occurring in individuals with the specific HLA class II allele DQB1*06:02. The autoimmune response leads to the selective destruction of orexin-producing neurons. The level of CSF orexin deficiency probably reflects the clinical spectrum of narcolepsy. Abbreviations: HLA, human leukocyte antigen; NT1, narcolepsy type 1; CA, cataplexy; w/o, without; NT2P, narcolepsy type 2 with HLA DQB1*06:02 positive; NT2N, narcolepsy type 2 with HLA DQB1*06:02 negative; CSF, cerebrospinal fluid.

**Table 1 ijerph-19-06285-t001:** Comparison of demographic variables among the types of narcolepsy spectrum disorders.

	NT1	NT2 [NT2P/NT2N]	IH
*n*	94	83 [36/47]	57
Mean age (SD)	50.7 (14.4) *^a,^ *^b^	45.8 (7.8) [45.6 (7.9)/45.9 (7.8)]	45.5 (9.6)
Sex: male (%)	56 (59.6)	45 (54.2) [13 (36.1) *^c^/32 (68.1)]	35 (63.8)
BMI (SD)	26.7 (4.4) *^a,^ *^b^	24.1 (2.1) [24.0 (1.6)/24.2 (2.5)]	24.1 (4.3)
BMI ≥ 25 (%)	54 (57.4) *^d^	21 (25.3) *^c^ [7 (19.4)/14 (29.8)]	18 (31.6)
OSA with AHI ≥ 15/h (%)	16 (17.0)	10 (12.0) [5 (13.9)/5 (10.6)]	9 (15.8)

Abbreviations: NT1—narcolepsy type 1; NT2—narcolepsy type 2; NT2P—narcolepsy type 2 with human leukocyte antigen DQB1*06:02 positive; NT2N—narcolepsy type 2 with human leukocyte antigen DQB1*06:02 negative; IH—idiopathic hypersomnia; SD—standard deviation; BMI—body mass index; OSA—obstructive sleep apnea; AHI—apnea–hypopnea index. *^a^ *p* < 0.05 vs. NT2 (one-way ANOVA, *p* value adjustment followed by Bonferroni method); *^b^ *p* < 0.05 vs. IH (one-way ANOVA, *p* value adjustment followed by Bonferroni method); *^c^ *p* < 0.05 (chi-square test, adjusted residual < −1.96); *^d^ *p* < 0.05 (chi-square test, adjusted residual > 1.96).

**Table 2 ijerph-19-06285-t002:** Comparison of complication prevalence among the types of narcolepsy spectrum disorders.

	NT1	NT2 [NT2P/NT2N]	IH
Hypertension (%)	46 (48.9) *^a^	22 (26.5) *^b^ [14 (38.9) *^a^/8 (17.0)]	18 (31.6)
Diabetes (%)	22 (23.4)	10 (12.0) [7 (19.4)/3 (6.4)]	6 (10.5)
Dyslipidemia (%)	24 (25.5)	17 (20.5) [13 (36.1) *^a^/4 (8.5) *^b^]	9 (15.8)
MRD (%)	55 (58.5) *^c^	33 (39.8) [20 (55.6) *^a^/13 (27.7)]	26 (45.6)

Abbreviations: NT1—narcolepsy type 1; NT2—narcolepsy type 2; NT2P—narcolepsy type 2 with human leukocyte antigen DQB1*06:02 positive; NT2N—narcolepsy type 2 with human leukocyte antigen DQB1*06:02 negative; IH—idiopathic hypersomnia; MRD—metabolic-syndrome-related disorder. *^a^ *p* < 0.05 (chi-square test, adjusted residual > 1.96); *^b^ *p* < 0.05 (chi-square test, adjusted residual < −1.96); *^c^ *p* < 0.05 (chi-square test, adjusted residual < 1.96)

**Table 3 ijerph-19-06285-t003:** Multiple logistic regression analysis of factors associated with MRD in NSD (*n* = 234).

	Odds Ratio	95% Confidence Interval	*p*-Value
Age	0.95	0.93–0.98	<0.01 *
Sex: male	1.47	0.82–2.64	0.20
BMI	0.85	0.78–0.93	<0.01 *
OSA with AHI ≥ 15/h	2.50	1.04–6.00	<0.05 *
Category of NSD			0.73
NT1 (ref. IH)	0.99	0.46–2.16	0.99
NT2 (ref. IH)	1.27	0.60–2.66	0.53

A forced entry method was used in this logistic regression analysis, focusing on clinically significant variables of NSD (age, sex, BMI, presence of OSA with AHI ≥ 15/h, and the category of NSD). All independent variables of this analysis are shown in this table, and statistically significant results are marked with *. Abbreviations: MRD—metabolic-syndrome-related disorder; NSD—narcolepsy spectrum disorders; BMI—body mass index; OSA—obstructive sleep apnea; AHI—apnea–hypopnea index; NT1—narcolepsy type 1; NT2—narcolepsy type 2; IH—idiopathic hypersomnia; ref—reference.

**Table 4 ijerph-19-06285-t004:** Multiple logistic regression analysis of factors associated with MRD in NT1 (*n* = 94).

	Odds Ratio	95% Confidence Interval	*p*-Value
Age	1.05	1.01–1.09	<0.01 *
Sex: male	1.16	0.45–3.01	0.77
BMI	1.13	1.01–1.27	<0.05 *
OSA with AHI ≥ 15/h	3.84	0.74–20.00	0.11

A forced entry method was used in this logistic regression analysis, focusing on clinically significant variables of NT1 (age, sex, BMI, and presence of OSA with AHI ≥ 15/h). All independent variables of this analysis are shown in this table, and statistically significant results are marked with *. Abbreviations: MRD—metabolic-syndrome-related disorder; NT1—narcolepsy type 1; BMI—body mass index; OSA—obstructive sleep apnea; AHI—apnea–hypopnea index.

**Table 5 ijerph-19-06285-t005:** Multiple logistic regression analysis of factors associated with MRD in NT2 (*n* = 83).

	Odds Ratio	95% Confidence Interval	*p*-Value
Age	1.03	0.97–1.10	0.32
Sex: male	0.51	0.18–1.45	0.21
BMI	1.49	1.09–2.02	<0.05 *
OSA with AHI ≥ 15/h	1.07	0.21–5.54	0.93
HLA DQB1*06:02 positivity	3.64	1.28–10.40	<0.05 *

A forced entry method was used in this logistic regression analysis, focusing on clinically significant variables of NT2 (age, sex, BMI, presence of OSA with AHI ≥ 15/h, and the presence of HLA DQB1*06:02 positivity). All independent variables of this analysis are shown in this table, and statistically significant results are marked with *. Abbreviations: MRD—metabolic-syndrome-related disorder; NT2—narcolepsy type 2; BMI—body mass index; OSA—obstructive sleep apnea; AHI—apnea–hypopnea index; HLA—human leukocyte antigen.

**Table 6 ijerph-19-06285-t006:** Multiple logistic regression analysis of factors associated with MRD in IH (*n* = 57).

	Odds Ratio	95% Confidence Interval	*p*-Value
Age	1.09	1.01–1.18	<0.05 *
Sex: male	0.53	0.15–1.90	0.33
BMI	1.17	0.99–1.38	0.07
OSA with AHI ≥ 15/h	2.42	0.43–13.60	0.32

A forced entry method was used in this logistic regression analysis, focusing on clinically significant variables of IH (age, sex, BMI, and presence of OSA with AHI ≥ 15/h). All independent variables of this analysis are shown in this table, and statistically significant results are marked with *. Abbreviations: MRD—metabolic-syndrome-related disorder; IH—idiopathic hypersomnia; BMI—body mass index; OSA—obstructive sleep apnea; AHI—apnea–hypopnea index.

## Data Availability

All data generated or analyzed during this study are available from the corresponding author upon reasonable request.

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
