# Peer review of "Metabolic-Syndrome-Related Comorbidities in Narcolepsy Spectrum Disorders: A Preliminary Cross-Sectional Study in Japan"

_ijerph, 2022, doi:10.3390/ijerph19106285_

Round 1

Reviewer 1 Report

This manuscript reports a study conducted on understanding the prevalence of metabolic syndrome-related disorders (MRD) in patients suffering from narcolepsy. Specifically, the authors chose patients suffering from NT1, NT2 and IH without LST from three different clinics. Subsequently, they investigated the prevalence of conditions such as diabetes, hypertension, obesity and BMR in these patients as readouts of MRD. They have also reported age, gender and HLA-DQB1*06:02 in these patients. The methods and results are well described, and the discussion is very well written, even highlighting the shortcomings that this study might have. The findings of this study will be of sufficient interest to the scientific community.
There are few minor edits that I would like to suggest:

  1. It will be useful if the authors also provide the statistics for the different metabolic parameters from the general population of the same age groups and compare them with those seen in narcolepsy patients.
  2. Since a large number of acronyms has been used in this manuscript, adding a table enlisting all the acronyms in this paper along with their full forms will be immensely helpful to the general audience.
  3. In line 49, add 'are' between 'NT2 and 'present'. 

Reviewer 2 Report

  1. The article title should mention the article type.
  2. The reviewer believes that the exclusion of individuals less than 35 years would significantly impact the results. It is advised to include this data and do a regression to exclude the confounding variable.
  3. What was the model used in the logistic regression?
  4. Who did the diagnosis of narcolepsy?
  5. How was the influence of the medications in use assessed?
  6. Could the authors explain the low number of IH subjects when compared to other groups? Could these group differences affect the analysis?
  7. How was calculated the power of the study?
  8. Please describe the data distribution.
  9. It is advised to mark significant results with *
  10. A figure about the main hypothesis of the study would greatly impact the results.

Reviewer 3 Report

I was very happy to read this nice work abut Metabolic syndrome in narcolepsy

First, for me the introduction is very well written, nothing to notice, it’s clear that the metabolic syndrome is a major concern in the context of narcolepsy.

Material and methods

2.1. Surveyed participants and procedures

During the diagnosis :

  • For the NT2 patients and hypersomniac patient, did you dispose of an actigraphic recording to exclude any case of bad sleep hygiene of severe and chronic sleep deprivation ?
  • For all, did they underwent an MRI ?

Line 74 : Sorry but I don’t understand the justification for selecting patient of > 35 y.o., we already see cases of metabolic syndrome in children with NT1 indicating that the metabolic state of this patients has to be considered even at a young age.

Line 114 : Could you justify more ? You only kept patients with IAH <5 and > 15/h but treated right? How much patients did you excluded according this criteria ? Was it to limit heterogeneity of the sample ? Or would you consider to keep this patients to see if moderate OSA is a factor favoring MSRD, maybe differently in your three groups?

2.2

Line 131 : We would need to have the definition of NT2P and NT2N because we only have it further (in the table 1 caption).

Results

It would be very interesting to have the age of diagnosis of the patients, in particular for the NT1 and the delay since the diagnosis. Is it a factor that could be associated with MRSD?

Also, you didn’t describe the severity, in term of somnolence maybe? All the patient are well controlled by their actual treatment? Is there a residual somnolence for some of them? Did you evaluate it? (Epworth Score?). Is it a factor that could be associated with MSRD?

Since the major effect you see is about hypertension, it is important to understand the role of the treatments. Do you have access to these data ? it has to be described, at least evaluate if the treatments are different between your 3 groups (different dosages ? different associations ? Number of simultaneous treatments?).  

Diabetes definition: It might be interesting to see if your patients have some insulin-resistance, using fasting insulin and glucose levels, HOMA index; it can highlight some abnormalities of glucose metabolism that would allow the clinician to start a specific medical care before reaching the stage of diabetes.

Thank you by advance for your answers.

Have a nice day

Author Response

Material and methods

2.1. Surveyed participants and procedures

During the diagnosis :

(Q1.)

For the NT2 patients and hypersomniac patient, did you dispose of an actigraphic recording to exclude any case of bad sleep hygiene of severe and chronic sleep deprivation ?

(Response)

Thank you for pointing this out. An actigraphic recording was not performed on all the subject patients. However, sleep hygiene instruction was given to subject patients prior to PSG and MSLT, with instructions to sleep longer (at least 7 hours per day), and they were asked to record sleep diaries for approximately one month to exclude insufficient sleep syndrome and circadian rhythm sleep-wake disorders. We added this description in the text.

<Revised manuscript> (in lines 80-83 of the redlined version manuscript)

Sleep hygiene instruction was given prior to PSG and MSLT, with instructions to sleep longer (at least 7 hours per day) and to record a sleep diary for approximately one month to exclude insufficient sleep syndrome and circadian rhythm sleep-wake disorders.

(Q2.)

  • For all, did they underwent an MRI ?

(Response)

Head MRI has not been performed on all the subjects. However, there were no patients who were suspected as having secondary narcolepsy due to head trauma, multiple sclerosis, or neurodegenerative diseases in this study.

(Q3.)

Line 74 : Sorry but I don’t understand the justification for selecting patient of > 35 y.o., we already see cases of metabolic syndrome in children with NT1 indicating that the metabolic state of this patients has to be considered even at a young age.

(Response)

Thank you for this insightful comment.

We selected the subjects aged > 35 years, since higher mortality rates have already been reported especially among elderly patients with narcolepsy in previous studies (Ohayon, Sleep 2014, doi:10.5665/sleep.3470; Jennum, Sleep medicine 2017, doi:10.1016/j.sleep.2017.03.029).

Furthermore, in former complication studies on more than 100 cases of narcolepsy as shown below, the mean ages were more than 40 years.

(1) N=320, 40.27 years (Ohayon,2013 United States, doi: 10.1016/j.sleep.2013.03.002)

(2) N=138, 48.44 years (Kok, 2003 Netherlands, DOI: 10.1038/oby.2003.156)

(3) N=132, 47.7 years (Dahmen, 2001 Germany, doi: 10.1007/s004060170057)

However, it is possible that CSF-orexin deficiency may cause obesity-related metabolic syndrome even at a relatively young age which we had described in the manuscript “the onset of NT1 is likely to lead to rapid weight gain, which could increase to the level of pathological obesity at a relatively young age” (in line 255-257 of the redlined version manuscript). So, we have revised the limitation section and the conclusion section to add that further studies in the future should include younger patients.

<Revised manuscript>

(Limitation section; in lines 332-334 of the redlined version manuscript)

Sixth, as we described in the method, we excluded patients aged < 35 years. However, CSF-orexin deficiency may cause metabolic syndrome at a relatively young age. Further studies of this kind should include younger patients.

(Conclusions section; in lines 356-359 of the redlined version manuscript)

Further studies with larger sample sizes including younger patients and general population as controls and prospective follow-up will be needed to confirm our results as well as to investigate the CSF orexin levels among patients with NT2.

(Q4.)

Line 114 : Could you justify more ? You only kept patients with IAH <5 and > 15/h but treated right? How much patients did you excluded according this criteria ? Was it to limit heterogeneity of the sample ? Or would you consider to keep this patients to see if moderate OSA is a factor favoring MSRD, maybe differently in your three groups?

 (Response)

Thank you for this important comment. Our wording might cause misleading.

We have made the following changes to the manuscript from ‘’We excluded patients with mild OSA (AHI < 15/h) from the analyses in this study’’ to ’’Patients with moderate to severe OSA were selected for analyses in this study’’(in line 120-121 of the redlined version manuscript)

There are 21 patients with mild OSA having AHI 5-15/hr (NT1, n=5; NT2P, n=4; NT2N, n=6; IH, n=6). They were included in this study, but were excluded from the analyses about the relationship between OSA and MRD. Among these patients having mild OSA, no patients were treated with nCPAP, since medical insurance coverage for nCPAP treatment is set to be limited to patients with AHI >20/hr in Japan. However, one patient having mild OSA was treated using oral appliance.

We selected only patients with moderate to severe OSA in the analyses of this study for following two reasons. First, the diagnosis of mild OSA seemed to be unclear, as we had mentioned in the text (in line 118-120 of the redlined version manuscript). Second, the association with the development of metabolic syndrome has been thought to be stronger in patients with moderate to severe OSA than in those with mild OSA (Shaoyong Xu, BMC Pulm Med. 2015. doi: 10.1186/s12890-015-0102-3.).

<Revised manuscript> (in line 120-123 of the redlined version manuscript)

Patients with moderate to severe OSA were selected for the analyses in this study, considering that all the participants with NSD had symptoms of EDS irrespective of whether they had OSA or not.

2.2

(Q5.)

Line 131 : We would need to have the definition of NT2P and NT2N because we only have it further (in the table 1 caption).

(Response)

Thank you for this practical comment. Definition of NT2P and NT2N were described in method (in line 90-92 of the redlined version manuscript). However, a large number of acronyms used in this manuscript. So, we have added the list of all abbreviations at the end of the text.

<Revised manuscript> (in line 389 of the redlined version manuscript)

Abbreviations:

AHI, apnea-hypopnea index

ANOVA, one way analysis of variance

BMI, body mass index

CSF, cerebrospinal fluid

EDS, excessive daytime sleepiness

HLA, human leukocyte antigen

ICSD-2, the International Classification of Sleep Disorders, Second Edition

ICSD-3, the International Classification of Sleep Disorders, Third Edition

IH, idiopathic hypersomnia

LST, long sleep time

MRD, metabolic syndrome-related disorder

MSLT, multiple sleep latency tests

nCPAP, nasal continuous positive airway pressure

NSD, narcolepsy spectrum disorder

NT1, narcolepsy type 1

NT2, narcolepsy type 2

NT2P, human leukocyte antigen DQB1*06:02 positive narcolepsy type 2

NT2N, human leukocyte antigen DQB1*06:02 negative narcolepsy type 2

OSA, obstructive sleep apnea

PSG, polysomnography

REM, rapid eye movement

SD, standard deviation

Results

(Q6.)

It would be very interesting to have the age of diagnosis of the patients, in particular for the NT1 and the delay since the diagnosis. Is it a factor that could be associated with MRSD?

(Response)

We could obtain the information about the onset age of hypersomnia from 214 out of 234 subject patients in this study. Onset age differed significantly among the three categories (p = 0.032, one-way ANOVA), and the age of NT2 looked to be younger than those with NT1 and IH. However, post-hoc Bonferroni analysis on the age showed no difference between NT2 and NT1 (p=0.054), and between NT2 and IH (p=0.116).

NT1

NT2

[NT2P/NT2N]

IH

n

78

81

 [35/46]

55

Mean age at onset (SD)

19.5 (10.1)

16.6 (2.3)

[17.1 (2.3)/16.2 (2.2)]

19.4 (9.0)

We also performed multiple logistic regression analysis with onset age as an independent variable. As a result, onset age did not show an association with the presence of MRD.

Multiple logistic regression analysis of factors associated with MRD in NSD (n = 234)

Odds ratio

95% confidence interval

P value

Age

1.06

1.02–1.10

< 0.01

Age at onset of hypersomnia

1.03

0.98-1.08

0.23

Sex: male

0.56

0.30–1.06

0.07

BMI

1.15

1.05–1.27

< 0.01

OSA with AHI ≥ 15/h

2.96

1.17–7.47

< 0.05

Category of NSD

0.73

NT1 (ref. IH)

0.97

0.42–2.24

0.94

NT2 (ref. IH)

0.81

0.37–1.76

0.59

(Q7.)

Also, you didn’t describe the severity, in term of somnolence maybe? All the patient are well controlled by their actual treatment? Is there a residual somnolence for some of them? Did you evaluate it? (Epworth Score?). Is it a factor that could be associated with MSRD?

(Response)

We are very sorry that we do not have the detailed information about the patients’ severity of EDS (such as Epworth score) both at the first visit to our clinic and at the survey. However, all of them were taking psychostimulants at the time of the survey, and it was impressed that the symptom of EDS were sufficiently controlled in most of the subjects. Given this, residual somnolence could not be a factor associated with the presence of MRD.

(Q8.)

Since the major effect you see is about hypertension, it is important to understand the role of the treatments. Do you have access to these data ? it has to be described, at least evaluate if the treatments are different between your 3 groups (different dosages ? different associations ? Number of simultaneous treatments?).  

(Response)

I’m sorry to say that we did not obtained the detailed data of psychostimulants, since it is impossible to convert to equivalent value of psychostimulants (modafinil, methylphenidate hydrochloride, pemoline). It seems that drug evaluation was often difficult in other previous studies. However, hypertension was the most frequently identified complication in this study. The effect of psychostimulants on blood pressure cannot be ignored. We performed additional analysis for the influence of the medications. We could confirmed psychostimulant drugs for 90 patients out of 234 patients.

modafinil single agent (n=43), pemoline single agent (n=39), methylphenidate hydrochloride single agent (n=3), combination of multiple-pshychostimulants (n=5),

The χ-square tests for the most common single drug use, modiodar dosage and the presence of hypertension, are as follows.

Hypertension (+)

Hypertension (-)

Modiodal 100mg (n=2)

1

1

Modiodal 200mg (n=31)

9

22

Modiodal 300mg (n=10)

6

4

As drug dosage increases, patients with hypertension appears to be increasing.

However, there were no statistically significant differences due to small numbers (p=0.145, Fisher’s exact test).

We would like to examine the relationship between complications and medication between the three groups (NT1, NT2, IH), but the small number of N makes this analysis difficult.

As we described in the limitation (in line 330-332 of the redlined version manuscript), this is one of the weak points of this study and should be resolved in future studies.

(Q9.)

Diabetes definition: It might be interesting to see if your patients have some insulin-resistance, using fasting insulin and glucose levels, HOMA index; it can highlight some abnormalities of glucose metabolism that would allow the clinician to start a specific medical care before reaching the stage of diabetes.

(Response)

Thank you for this insightful comment. In this study, the diagnosis of diabetes were made with the hemoglobin A1c according to the diagnostic criteria for diabetes mellitus in Japan. By using this, there was no significant difference in the incidence of diabetes by a narrow margin (p = 0.05) among the three group. As you pointed out, abnormalities of glucose metabolism (insulin-resistance) may appear before reaching to the clinical stage of diabetes. We added this description in limitation.

<Revised manuscript> (in lines 335-339 of the redlined version manuscript)

Seventh, as we mentioned in the result, there was no significant difference in the incidence of diabetes by the narrow margin (p = 0.05) among the three NSD group. However, abnormalities of glucose metabolism may be shown before reaching the stage of diabetes. It might be effective to evaluate not only hemoglobin A1c but also insulin-resistance (such as HOMA index) in the further studies.

Round 2

Reviewer 2 Report

I would like to congratulate the authors on the great improvement of the manuscript.

1) The article title should mention the article type. ‘‘a preliminary study in Japan’’. ‘‘cross-sectional study’’

2) The authors should describe the variables of clinical significance used in the regression.

Author Response

We really appreciate your review. Your peer review helped us to complete this paper. The responses to all comments have been prepared with attached word file. Thank you.

Reviewer 3 Report

Thank you for this new version.

You answered all my questions, thank you.

Author Response

We really appreciate your review. Your peer review helped us to complete this paper. Thank you very much.